# The Vestibular Nuclei: A Cerebral Reservoir of Stem Cells Involved in Balance Function in Normal and Pathological Conditions

**DOI:** 10.3390/ijms25031422

**Published:** 2024-01-24

**Authors:** Guillaume Rastoldo, Brahim Tighilet

**Affiliations:** 1Aix Marseille Université-CNRS, Laboratoire de Neurosciences Cognitives, LNC UMR 7291, 13331 Marseille, France; guillaume.rastoldo@inserm.fr; 2GDR Vertige CNRS Unité GDR2074, 13331 Marseille, France

**Keywords:** neurogenesis, gliogenesis, neural stem cells, neural stem cells niche, vestibular nuclei, vestibular compensation, balance

## Abstract

In this review, we explore the intriguing realm of neurogenesis in the vestibular nuclei—a critical brainstem region governing balance and spatial orientation. We retrace almost 20 years of research into vestibular neurogenesis, from its discovery in the feline model in 2007 to the recent discovery of a vestibular neural stem cell niche. We explore the reasons why neurogenesis is important in the vestibular nuclei and the triggers for activating the vestibular neurogenic niche. We develop the symbiotic relationship between neurogenesis and gliogenesis to promote vestibular compensation. Finally, we examine the potential impact of reactive neurogenesis on vestibular compensation, highlighting its role in restoring balance through various mechanisms.

## 1. Introduction

Neurogenesis is a fascinating process engaging the community within the field of neuroscience for more than 100 years and refers to the generation of new neurons in the brain. Once believed to be restricted to early developmental stages, recent discoveries have illuminated the existence of neurogenesis in specific regions of the adult mammalian brain, challenging traditional dogmas and opening new research areas [1].

Spontaneous neurogenesis represents the ongoing generation of neurons in defined neurogenic niches, contributing to the structural and functional plasticity of the adult brain. Conversely, reactive neurogenesis is triggered in response to injury, neurodegenerative disorders, or other pathological conditions and could serve as a reparative mechanism. Understanding the molecular and cellular dynamics that govern the various facets of neurogenesis is essential to unraveling the complexities of neuronal regeneration through the use of neural stem cell therapies, ultimately leading to the design of effective therapeutic applications adapted to brain pathologies [2,3,4].

The discovery of neural stem cells has actually revolutionized our understanding of brain plasticity, enabling us to better decipher the intimate mechanisms of brain development and the cellular and molecular events correlated with neuropathologies. These cells are primarily located in specialized regions of the brain, called neurogenic niches, such as the subgranular zone of the dentate gyrus and the subventricular zone of the lateral ventricles. However, other brain regions such as the hypothalamus [5,6,7] and the vestibular nuclei [8] have subsequently been discovered with a pool of neural stem cells.

In this review, we focus on neurogenesis in the vestibular nuclei and, through this prism, explore brain plasticity, regeneration and its therapeutic potential. From neural stem cells to spontaneous or reactive neurogenesis, we will delve into the potential activation mechanisms, functional relevance, and current debates surrounding vestibular neurogenesis. We retrace almost 20 years of research into vestibular neurogenesis, from its discovery in the feline model in 2007 to the recent discovery of a pool of neural stem cells within vestibular nuclei in rats.

## 2. Search Strategy and Selection Criteria

We have performed a Pubmed literature search of articles with search terms including “neurogenesis”, “neural stem cells”, “cellular proliferation”, “neurogenic niche”, “vestibular compensation”, “vestibular injury”, “vestibular damage”, “labyrinthectomy”, “vestibular neurectomy” and “vestibular nucleus”. Selection criteria included recent articles (2013–2023) on vestibular plasticity and functional recovery after vestibular injury mainly focusing on unilateral vestibular neurectomy. In addition, we include pertinent articles on the subject published earlier (<2013) where appropriate.

## 3. Neural Stem Cells and Neurogenic Niche

Neural stem cells (NSC) are undifferentiated cells capable of multiplying identically and generating specialized cells through cellular differentiation. NSCs are localized in specialized cellular microenvironments called niches. Niches play a central role in integrating cellular signals from the environment and the needs of the organism to maintain the survival of the stem cells they host, as well as their proliferation, survival and differentiation [9,10]. By synthesizing data from several studies, we can suggest a hypothetical list of criteria attributing stem cell niche status to a nerve structure [11,12]. Thus, a niche consists of the following:–Neural stem cells.–Support cells that interact directly with stem cells and with each other via membrane receptors, gap junctions and soluble factors (growth factors, cytokines and various proteins; for review, see [11]). For neurogenic niches, the supporting cells are vascular cells, astrocytes, microglial cells, pericytes and ependymal cells.–Blood vessels that transport oxygen and nutrients. Vascularization also enables the recruitment of inflammatory cells and other circulating cells into the niche, as well as the entry and exit of stem cells. We can also highlight the role of blood vessels in the choroid plexus in regulating cerebrospinal fluid production and composition (for review, see Karakatsani et al., 2023 [13]).–An extracellular matrix that provides structure, organization and mechanical signals to the niche.–Nerve fibers that may also communicate physiological signals to the niche.

While not every niche necessarily integrates all these distinct elements, it is clear that a neurogenic niche represents a complex, dynamic entity in which the integration of multiple signals enables precise control of stem cell proliferation and survival. The two classical niches, the subventricular zone of the lateral ventricles (SVZ) and the subgranular zone of the dentate gyrus (SGZ), share key elements of a neurogenic niche despite being different [14]. NSCs are therefore multipotent cells capable of self-renewal, thus maintaining the stem cell pool throughout life, and can also generate neural progenitors capable of proliferating and acquiring different cellular phenotypes (astrocytes, oligodendrocytes or neurons).

## 4. Spontaneous Neurogenesis

Neurogenesis is defined as the brain process leading to the genesis of functional neurons from neural stem cells (NSCs). There are two main zones of spontaneous neurogenesis in the adult brain of different mammalian species, including rodents [15]: the subventricular zone (SVZ) bordering the lateral ventricles and the subgranular zone (SGZ) of the dentate gyrus of the hippocampus. The SVZ is the most abundant neurogenic zone in the central nervous system, supplying the olfactory bulb with several thousand newly generated interneurons every day via the rostral migratory flow. This constant flow of new neurons appears to be essential for odor discrimination, learning and memory tasks in rodents [16]. Thousands of neurons are generated every day in rodents, but approximately 30–70% do not survive and only a small proportion integrate permanently into the neural circuit [16,17].

Spontaneous adult neurogenesis in the hippocampus is restricted to the SGZ of the dentate gyrus. Neural stem cells in this area generate neural progenitors, which in turn differentiate into neuroblasts, embryonic nerve cells that give rise to neurons. These neuroblasts then develop apical dendritic extensions and extend their axons towards Ammon’s horn 3 (CA3), forming mossy fibers. The maturation of neuroblasts into functional neurons integrated into pre-existing hippocampal neural networks is a long process lasting several weeks [18]. Maturation of young neurons is considered to be achieved around 2 months after birth. The structural and physiological properties of newly generated neurons are then similar to those of neighboring mature granule cells. It would appear that adult hippocampal neurogenesis has a major impact on learning, spatial navigation, long-term memory, the removal of memory traces from the hippocampus and the reorganization of memory towards extra-hippocampal substrates [19].

It is important to note that adult spontaneous neurogenesis is not limited exclusively to the SVZ and SGZ (for review: [20,21,22]). Indeed, spontaneous new neuron production has been described in the striatum [23,24], neocortex [24,25], substantia nigra [26], piriform cortex [27], amygdala [27], inferior temporal cortex [27], dorsal vagal complex [28], caudate nucleus [29], olfactory tubercles [30], circumventricular organs [31] and hypothalamus [7,32,33] (Figure 1).

## 5. Reactive Neurogenesis

While spontaneous adult neurogenesis under physiological conditions produces mature neurons that migrate, differentiate and then integrate into functional neural networks, neurogenesis under pathological conditions (or reactive neurogenesis) raises many questions, notably as to its beneficial or deleterious role. Moreover, different pathologies or lesions of the central nervous system can stimulate neurogenesis in the classic niches (ZSV and ZSG) but also in other brain regions (cerebral cortex, striatum, spinal cord, vestibular nuclei, etc.). Thus, under the influence of specific signals, a tissue previously considered non-neurogenic can free itself from endogenous restrictions and allow time-limited neurogenesis to renew its stock of neurons in the damaged area. This is known as reactional or secondary neurogenesis, as opposed to spontaneous or primary neurogenesis. Apart from ZSV and ZSG, there are two possibilities to explain the neurogenesis observed in a damaged region: either the brain structure harbors NSCs that will be able to produce new neurons, or the new neurons originate from classical niches (essentially ZSV) via different pathways (along blood vessels or via CSF) [34,35].

## 6. Vestibular Neurogenesis

The existence of reactive neurogenesis restricted to deafferented vestibular nuclei after unilateral vestibular neurectomy (UVN) was demonstrated for the first time by our laboratory in 2007 [36]. Using a cell division marker, bromodeoxyuridine (BrdU) in UVN-cats, we demonstrated strong cell proliferation exclusively in all deafferented vestibular nuclei, with peak expression 3 days after injury. Interestingly, one month after injury, 70% of these newly generated cells observed in deafferented vestibular nuclei survived and differentiated into glial cells (astrocyte and microglia) and mature GABAergic neurons. The question following this remarkable discovery was whether this process of neurogenesis contributed to vestibular compensation or was simply an aberrant artifact. To answer this crucial question, we showed that a 30-day post-injury infusion of the anti-mitotic cytosine-β-D-arabinofuranoside (AraC) in the fourth ventricle of animals in vivo completely blocked cell proliferation and, consequently, neurogenesis. From a behavioral approach, after vestibular injury and AraC infusion, postural and locomotor recovery was considerably delayed in cats. However, no alteration in the restoration of spontaneous horizontal nystagmus was observed [37]. This novelty represents a new dogma in vestibular neurophysiology. Conversely, continuous infusion of BDNF from day one post-NVU dramatically increases rates of cell proliferation, survival and differentiation without affecting the choice of neuronal or glial fate of newly formed cells [38]. Under these conditions of chronic BDNF treatment, animals significantly recovered balance and posture earlier. Conversely, blocking BDNF-TrkB signaling with a blocking agent, K252a, significantly reduced reactive vestibular neurogenesis and prevented posturo-locomotor recovery in vestibulo-lesioned animals.

Many questions remain as to the establishment and relevance of reactive vestibular neurogenesis. Firstly, how does a mature brain environment, not listed as a neurogenic site, allow the genesis of new neurons in the post-injury period? One possibility is that these newly generated neurons migrate from established neurogenic niches, such as the SVZ, and then integrate into the vestibular nuclei bordering the fourth ventricle. Another option is that, under normal conditions, vestibular nuclei constitute a neurogenic niche harboring quiescent neural progenitors, as recently demonstrated [8,39]. Secondly, how is vestibular neurogenesis triggered and what is its functional significance?

## 7. A New Neurogenic Niche in the Brainstem: The Vestibular Nuclei

We provide the first demonstration of the presence of quiescent neural stem cells in vestibular nuclei of adult rats under physiological conditions [8]. In contrast to the high mitotic activity found in classical neurogenic niches, we observed few dividing NSCs in vestibular nuclei under physiological conditions [39], which might suggest a quiescent state of these cells. However, we have demonstrated activation of the vestibular niche with a strong increase in cell proliferation exclusively in deafferented vestibular nuclei following unilateral and irreversible loss of primary vestibular afferents by unilateral vestibular neurectomy [39]. Finally, newly generated cells differentiated into astrocytes, oligodendrocytes and neurons in adult rats subjected to UVN. Furthermore, in addition to this evidence, we have provided a list of arguments for legitimately assigning vestibular nuclei the status of a neurogenic niche (for more details, see [8]):

Firstly, we find, as in SGZ and SVZ, key components of the niche extracellular matrix (ECM) within vestibular nuclei. Specifically, chondroitin sulfate proteoglycans, hyaluronan and fractones [40,41,42,43] are highly expressed within vestibular nuclei [44,45] and along the walls of the IV ventricle (adjacent to the medial vestibular nucleus) [46].

Secondly, vascularization in the adult SVZ and SGZ is very dense and highly organized compared with non-neurogenic brain regions [13,47,48,49]. Of all brainstem structures, however, the medial vestibular nucleus is the most sensitive to ischemia, attesting to its extensive vascularization [50]. We found no other information in the literature concerning the other nuclei of the vestibular complex. However, unpublished observations by our team also seem to indicate that vascularization (after counting the number of blood vessels) is greater in the deafferented vestibular nuclei of the cat after NVU, underlining reactive angiogenesis to the vestibular lesion.

Thirdly, there is a privileged established relationship between the vestibular nuclei complex and the hippocampus. Indeed, surprisingly, there is no significant difference between the number of neural stem cells in the medial vestibular nucleus and the SGZ considered to be the neural stem cell reservoir of the hippocampus [8] (Figure 2). Peripheral vestibular lesions impair performance on learning and memory tasks related to the spatial environment [51]. Our group has just shown that unilateral vestibular neurectomy disrupts all areas of spatial memory, from working memory to reference memory and on-site object recognition. These deficits are associated with a lasting impairment of plasticity markers in the ipsilesional hippocampus. Indeed, unilateral irreversible vestibular loss performed in the adult rat induces a significant reduction in cell proliferation in the ipsilateral dentate gyrus of the dorsal hippocampus as well as a significant increase in GluN2B subunit of NMDA receptors in the ipsilateral CA3 [52]. These results underline the crucial role of symmetrical vestibular information in the process of hippocampal neurogenesis and spatial memory management. These data contribute to a better understanding of the cognitive disorders observed in vestibular patients.

In addition, neurons in the medial vestibular nucleus, like those in the hippocampus, also exhibit synaptic plasticity phenomena such as LTP (long-term potentiation) and LTD (long-term depression) [53,54]. Both mechanisms are impaired in the medial vestibular nucleus after vestibular damage [54]. It is interesting to make the analogy with the results of our recent study using a video-tracking device [55]. At a behavioral level, we observed the alteration and recovery of certain parameters over a critical 7-day period following unilateral vestibular loss. This acute posturolocomotor phenotype is reminiscent of the developmental strategies used for gait acquisition. This suggests that compensation of locomotion and postural balance in vestibulo-impaired rats is a form of sensorimotor relearning combining specific cellular mechanisms in the vestibular nuclei and behavioral strategies. We can therefore assume that the LTP and LTD phenomena present in the vestibular nuclei are involved in the relearning of walking and posture after acute unilateral vestibular loss.

## 8. Why Neurogenesis Is Required in the Vestibular Nuclei

The question we need to ask is why vestibular nuclei contain a pool of neural stem cells. We can argue this question from a phylogenetic evolutionary point of view. Firstly, the peripheral vestibular system is one of the phylogenetically oldest structures [56]. Indeed, gravity sensing appears to be essential for complex life forms. Even plants have developed multiple mechanisms to detect the gravity that determines their orientation and final shape. In the animal kingdom, ‘simple’ life forms such as jellyfish possess sensory cells capable of detecting gravity and ocean currents [57]. As a reminder, jellyfish appeared 635 million years ago, yet the structural organization of sensory cells capable of detecting gravity remains similar to the human peripheral vestibular system. Indeed, the statocysts of scyphozoan jellyfish contain several thousand crystals (statoliths) made of bassanite [58], strongly evoking the organization of otolith organs in humans [59,60] (Figure 3). However, statoliths are located each in a cell called statocyst while human otoconia are located in an extracellular matrix called otoconial membrane.

The mammalian vestibular system as we know it today, with its 3 semicircular canals and otolith organs, appears to have appeared around 400 million years ago [56]. The purpose of the vestibular system, as it has evolved nowadays, is to provide information on gravity, balance (static and dynamic) and body position in space. A sudden unilateral loss of vestibular information causes numerous disorders in mammals, with a characteristic vestibular syndrome. If we once again take an evolutionary perspective, a loss of balance seriously compromises the survival of the species. The most basic evolutionary survival instinct is the fight–flight response. Essentially, it consists of physiological stress reactions designed to help the organism fight or flee in response to danger. Let us consider the case where there is no possible compensation for unilateral vestibular loss, as occurs with unilateral loss of vision or hearing. In this context, unilateral blindness or deafness remains disabling for the animal, but the impact on its survival cannot be placed at the same level as a unilateral vestibular loss without compensation. An animal unable to orientate itself in space, stabilize the image during head movement and maintain postural balance cannot survive. It is therefore vital to compensate for the unilateral loss of vestibular information. We can assume that quiescent neural stem cells present in the vestibular nuclei would be able to react rapidly in the event of loss of vestibular modality and thus ensure restoration of the posturolocomotor functions necessary for the survival of the species.

## 9. What Conditions Are Necessary to Activate the Vestibular Niche?

Unilateral vestibular neurectomy (UVN) can be a trigger for activating quiescent vestibular stem cells. Studies have shown that GABAergic signaling via GABA_A_ receptors maintains a quiescent state of NSCs by inhibiting their entry into the cell cycle, or restoring quiescence to activated NSCs [61,62,63,64]. We know that vestibular nuclei are continuously subjected to GABAergic inhibition via commissural vestibular pathways and cerebellovestibular pathways [65,66]. We can speculate that these two pathways are involved in maintaining the quiescent state of vestibular NSCs. The contralesional commissural inhibitory responses are enhanced after unilateral vestibular damage, and a significant release of GABA is observed in the deafferented vestibular nucleus [18]. However, despite the enhanced presence of GABA in the deafferented vestibular environment, NSCs leave their quiescent state indicating that other factors must be involved.

It is important to consider that the vestibular nuclei environment after UVN or vestibular damage is completely altered. In physiological conditions, a large number of factors released by the ECM, glia, neurons and blood factors regulate the balance between the quiescent and activated states of neural stem cells [67,68]. In pathological situations, unilateral labyrinthectomy (LU: irreversible destruction of peripheral vestibular receptors) or an injection of tetrodotoxin (toxin causing reversible blockade of vestibular nerve electrical activity) does not induce cell proliferation in deafferented vestibular nuclei in cats [69] indicating that neurogenesis within the vestibular nuclei is highly dependent on vestibular etiology at least in the cat. However, in rats, vestibular galvanic stimulation following UL triggers cell proliferation in the deafferented vestibular nuclei [70], whereas UL alone is not sufficient to activate the vestibular neurogenic niche. Unfortunately, the phenotype of the newly generated cells has not been determined; therefore, we cannot conclude the presence of new neurons. Another study in rats demonstrated vestibular neurogenesis with new GABAergic neurons within the deafferented vestibular nuclei following UL alone [71]; however, the results should be taken with caution. How can we explain these variations in cell proliferation between different models of vestibular lesions?

UVN is characterized by a strong glial and inflammatory response, as opposed to UL and tetrodotoxin injection. The exact role of inflammation on neural stem cells is controversial, as both deleterious and beneficial effects can be attributed to inflammation [72,73], suggesting that interactions between inflammation and NSCs are context dependent. An in vivo study demonstrates that after ischemic brain injury in the subventricular zone, the release of interferon-gamma (IFN-γ) at the site of injury enables the activation of selected NSCs [74]. Interestingly, in different models of unilateral vestibular deafferentations (mechanical or chemical UL by arsanilate) up-regulation of TNFα [75] and IL-1β [76] have been reported. No studies have quantified IFNγ release; however, Liberge and colleagues demonstrate up-regulation of the enzyme Manganese Superoxide Dismutase (MNSOD), which is itself up-regulated by TNFα, IL-1β and IFN-γ. It is therefore plausible that these various cytokines, massively released after UVN, participate in the activation of quiescent NSCs in the vestibular nuclei. Moreover, recent data support the role of endogenous inflammation following vestibular damage, in regulating neurogenesis and vestibular compensation. Indeed, pharmacological blockade or reinforcement of inflammation during the acute phase of vestibular compensation is accompanied by a worsening of the vestibular syndrome and a delay in vestibular compensation correlated with impaired vestibular neurogenesis [77,78]. However, more studies are needed to confirm these results but also to validate these findings in other models of vestibular damage.

Another interesting point concerns BDNF and its receptor, TrkB. Neurons immunoreactive to BDNF and TrkB increase transiently in deafferented vestibular nuclei, with peak expression at 3 days, concomitant with the peak in cell proliferation [79]. Furthermore, blockade of BDNF-TrkB signaling significantly reduces the cell proliferation observed in deafferented vestibular nuclei [38,71], highlighting the importance of BDNF in awakening vestibular NSCs. In conclusion, inflammation and BDNF would be elements of the deafferented vestibular environment contributing to the activation of the vestibular neurogenic niche after NVU. However, the molecular mechanisms involved in the awakening of these dormant cells remain to be elucidated.

While vestibular neurogenesis has not yet been confirmed in humans, it may be a key mechanism in the vestibular compensation process, leading to spontaneous recovery of postural and locomotor balance. Future studies related to the understanding of adult vestibular neurogenesis should provide therapeutic solutions to stimulate this process in order to improve postural and locomotor restoration in vestibular pathology.

## 10. Neurogenesis or Gliogenesis to Promote Vestibular Compensation?

While blocking cell proliferation delays vestibular compensation [37], it is intriguing to note that when pharmacological compounds [80] or a rehabilitation protocol [81] promote vestibular compensation, the number of newly formed neurons decreases in favor of increased microglial differentiation. Around 20% of neurons are newly generated with UVN alone, whereas with locomotor rehabilitation or thyroxine injections, which facilitate vestibular compensation, we observe an increase in cell proliferation but only 2 to 5% of new neurons. Meanwhile, microglial cell differentiation increases from 20% with UVN alone to 40–60%, representing the principal cell differentiation phenotype. It would seem that there are two different strategies for compensating for vestibular loss, depending on whether we help the system or leave it to fend for itself. The question now arises concerning why cell differentiation is driven towards predominantly microglial differentiation when vestibular compensation is improved.

After unilateral vestibular deafferentation, the goal of vestibular compensation is to rebalance the electrophysiologic asymmetry that is present between the vestibular nuclei on either side of the brainstem. An interesting recent study has shown that microglia can counteract neuronal hypoactivity during general anesthesia by shielding axo-somatic GABAergic inhibitory inputs and thus promoting neuronal activity [82]. This study is part of a more general context indicating that microglia can both negatively and positively regulate neuronal activity [83]. Since immediately after unilateral deafferentation, GABA release in the ipsilesional medial vestibular nucleus is strongly increased via the reciprocal commissural inhibitory system [84], microglia could play an important role here by blocking GABAergic inhibition and promoting a return to homeostatic neuronal activity.

Furthermore, we formulated a hypothesis in 2016 involving impaired GABAergic neurotransmission in the deafferented vestibular nuclei [38]. Indeed, after UVN we observed a downregulation of KCC2, a cation–chloride cotransporter that plays a crucial role in regulating intracellular chloride concentration in neurons. In the context of pathology, reduced expression of KCC2 disrupts GABA signaling, ultimately resulting in neuronal hyperexcitability [85,86,87,88]. KCC2 downregulation is closely related to activation of BDNF-TrkB signaling pathways by BDNF release from activated microglia [89]. Similar phenomena were observed in the UL model, with an increase in BDNF and TrkB expression, and a reduction in KCC2 expression in the ipsi-lesional MVN 8 h after UL [90]. Consequently, the specific downregulation of KCC2 induced by microglia in the deafferented vestibular nuclei may also restore the electrophysiological balance on both sides of the brainstem and participate in vestibular compensation.

In addition to central inflammation within the deafferented vestibular nuclei due to vestibular deafferentation, active microglia also respond to sensory deprivation. Indeed, by unilaterally trimming all the whiskers of mice, Kalambogias et al. demonstrated an active state of microglia within the barrel cortex [91]. They speculate that active microglia contribute to the structural plasticity of the barrel cortex induced by somatosensory deprivation. Following the unilateral loss of vestibular information, we can assume that the same mechanisms are used by active microglia to support the structural and synaptic rearrangements of vestibular compensation. Along the same lines, microglia are capable to strengthen certain synapses and eliminate others depending on sensory experience [92,93]. Therefore, newly generated microglia following UVN can shape vestibular microcircuitry and strengthen sensory afferents converging to vestibular nuclei to promote sensory substitution. Sensory substitution in the vestibular nuclei has already been demonstrated in monkeys, with the unmasking of extra-vestibular proprioceptive signals during compensation. Indeed, proprioceptive input on neurons in the vestibular nuclei that were silent before the lesion becomes sensitive to neck stimulation after unilateral labyrinthectomy. These proprioceptive signals, which are up-weighted during compensation, act synergistically with the remaining vestibular input to enhance the detection of head rotations [94,95,96,97].

## 11. What about Oligodendrogenesis Playing a Role in Vestibular Compensation?

In contrast, the role of oligodendrocytes in vestibular compensation is much less documented. There is only one study demonstrating up-regulation after UL of microRNA-219 [98], which is involved in oligodendrocyte differentiation and myelination [99]. Furthermore, administration of a microRNA-219-blocking oligomer delays vestibular compensation in UL rats [98]. We are the first to demonstrate an increase in oligodendrocytes in deafferented vestibular nuclei after UVN and the presence of oligodendrogenesis [39]. Oligodendrocytes are classically known as the myelinating cells of the central nervous system. They play an important role in increasing action potential conduction velocity by forming myelin sheaths, thus promoting saltatory conduction along the axon. The reactive oligodendrogenesis observed in deafferented vestibular nuclei could accelerate the myelination of axons from neoformed neurons. This process would facilitate communication within the local vestibular network, thus contributing to their functional integration. It cannot be ruled out that the increase in oligodendrocytes in vestibular nuclei induced by UVN may myelinate or increase the myelination of axons from visual and proprioceptive afferents. This process would enhance and facilitate the conduction of information from these two sensory modalities, thus promoting the sensory substitution phenomenon that is important for vestibular compensation. In recent years, oligodendrocytes have also been recognized as an important source of trophic factors, including BDNF, enhancing the survival, development and function of neighboring neurons [100,101,102]. Knowing the importance of BDNF in the development and maturation of the vestibular system [103,104], but also in vestibular compensation [38,105], increasing oligodendrocytes in deafferented vestibular nuclei would be of considerable functional benefit. Similarly, a recent review points to interactions between oligodendrocytes and astrocytes that can regulate inflammation in normal or pathological conditions [106]. Thus, the release of trophic factors by oligodendrocytes could also favor plasticity mechanisms within vestibular nuclei and promote vestibular compensation. Finally, it can be postulated that oligodendrogenesis and myelination are necessary to enhance motor learning and motor skill performance [107,108] and we have seen that compensation for vestibular behavioral deficits is associated with sensorimotor relearning [55].

## 12. What Contribution Does Reactive Neurogenesis Exert on Vestibular Compensation?

In the feline model, blocking cell proliferation following a 30-day post-injury infusion of an anti-mitotic agent (AraC) into the fourth ventricle significantly delayed recovery of the animals’ posturo-locomotor functions [37]. Newly formed neurons in the dentate gyrus of the hippocampus develop an elaborate dendritic arborization and a morphology similar to a mature neuron after 21 days [109]. In addition, 4 to 8 weeks are required for the neoformed neuron to integrate into the functional hippocampal network [110]. Around 4 weeks are required for cats subjected to UVN to restore postural–locomotor performance. This time constant suggests that full compensation of postural–locomotor deficits involves the maturation of new neurons and their integration into functional neural networks of deafferented vestibular nuclei. In the UVN rodent model, all vestibular tests show that rats compensate for their deficits in around 7 days [39,55,81,111,112]. Electrophysiological studies, albeit few in rodents after vestibular damage, support our behavioral data. The time constant for restoration of electrophysiological equilibrium after vestibular loss in all vestibular nuclei is around 7 days in guinea pigs [113,114,115,116]. For the median vestibular nucleus, restoration of spontaneous activity occurs between 2 and 3 days [117,118]. It would have been interesting to have more electrophysiological data from the lateral vestibular nucleus [116] since this nucleus is specifically dedicated to posturolocomotor function. As in the feline model, the question arises as to the functional role of neurogenesis in vestibular compensation in the rodent.

Initially, it would be interesting to block neurogenesis with an antimitotic and observe the functional consequences. Given the similar anatomical and functional organization of the vestibular system in these two species, we would expect a similar result to that obtained in the feline model. Despite the fact that functional integration of new neurons in the dentate gyrus takes between 4 and 8 weeks, no studies have been carried out to report the maturation and functional integration of neoformed neurons in deafferented vestibular nuclei. The intense cell proliferation induced by UVN produces both neural and glial cells. Indeed, UVN induces a strong glial response, with increased numbers of astrocytes, oligodendrocytes and microglial cells, within the deafferented vestibular nuclei. Functional maturation of astrocytes [119] is known to be faster than functional integration of neurons (2–3 weeks for astrocytes vs. 4–8 weeks for neurons). Similarly, remyelination is observed 3 weeks after oligodendrocyte progenitor cell transplantation [120,121]. We note that the maturation time constant of astrocytes and oligodendrocytes is longer than the functional restoration of vestibular deficits in rats (2–3 weeks vs. 1 week). This lack of correlation may indicate that neurogliogenesis is an important process for compensation but probably acts in concert with a multitude of other mechanisms [79]. Indeed, changes in gene expression, neuroinflammation, neurotrophins, hormonal systems, neurotransmitters, glial reaction and intrinsic properties of vestibular nuclei [75,122,123,124,125,126,127,128] appear early during vestibular compensation. Given the longer time constant for functional maturation of the glia and newly generated neurons, we can assume that these two phenomena would come later to reinforce and maintain vestibular compensation in the long term. Not all of the above-mentioned phenomena should be considered independently, but rather as a mosaic of coordinated events with different kinetics of action designed to contribute to the restoration of vestibular function.

The cellular and molecular environment of the deafferented vestibular nuclei is different from what it was before the lesion. The loss of vestibular information induces a major remodeling within the vestibular nuclei but also in various neural networks of the central nervous system [129,130]. This new equilibrium within the central vestibular network is not there to repair and mimic vestibular function as it was before the lesion. The system has found solutions and adapted to compensate for the loss of vestibular information to enable the animal to balance and be able to walk again.

## 13. Conclusions

The study of vestibular neurogenesis is still in its early stages, but it holds great promise for understanding vestibular physiopathology and improving the treatment of vestibular disorders. Morphological and electrophysiological studies would provide crucial insights into the maturation and functional integration of newly formed neurons during vestibular compensation. Similarly, characterizing vestibular stem cells and being able to determine the intra- and extracellular signals that activate these stem cells would shed light on the laws governing this new niche in the adult brain. In conclusion, future research is needed to better characterize vestibular neurogenesis, identify the factors that regulate this process and ultimately develop therapies that impact newly generated vestibular neurons with the view to restoring balance.

## Figures and Tables

**Figure 1 ijms-25-01422-f001:**
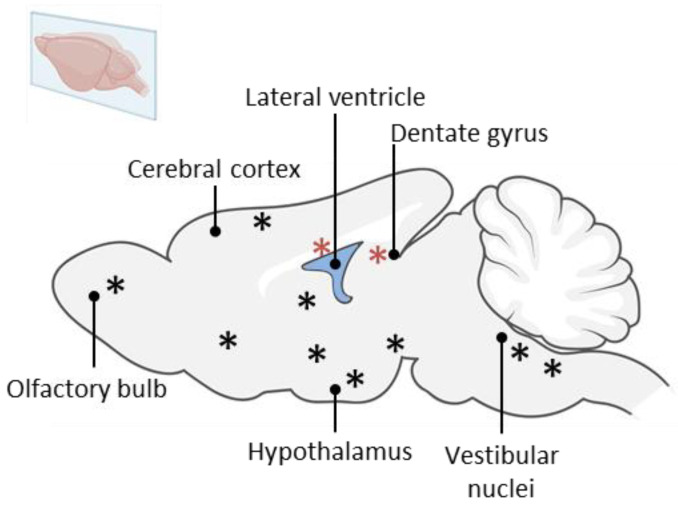
Graphical representation of the different spontaneous neurogenic regions in the adult rodent brain. Classical NSC niches, i.e., SVZ and SGZ are represented with a red asterisk. The striatum, the cortex, the amygdala, the dorsal vagal complex, the caudate nucleus, the olfactory tubercles, the hypothalamus and the vestibular nuclei are represented with black asterisk.

**Figure 2 ijms-25-01422-f002:**
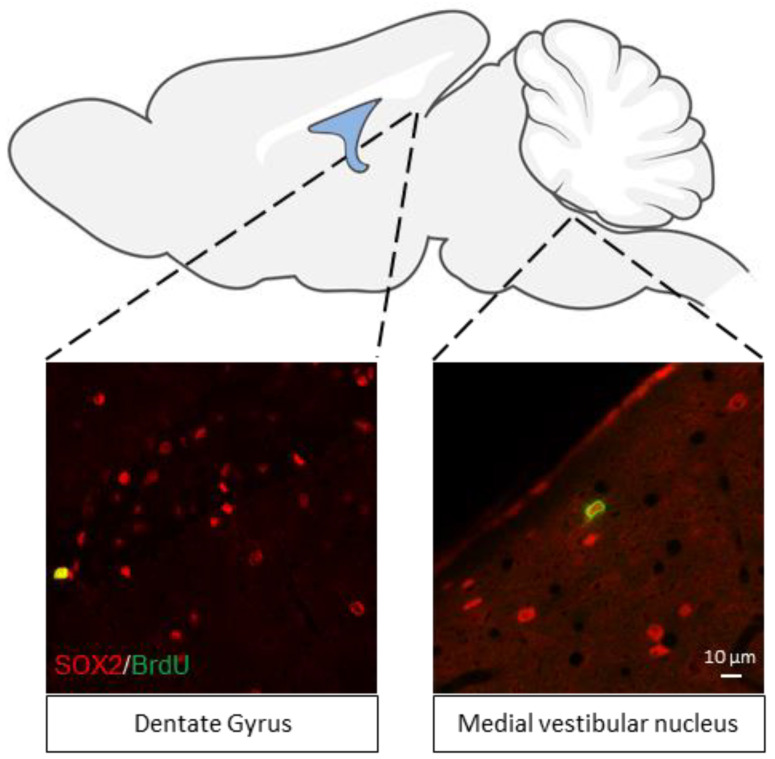
Graphical representation of the rodent brain with a particular focus on the classical neurogenic niche SGZ of the dentate gyrus and the medial vestibular nucleus (MVN). Proliferative neural stem cells (SOX2+/BrdU+) and quiescent neural stem cells are found in both the SGZ and the MVN. Adapted from [8].

**Figure 3 ijms-25-01422-f003:**
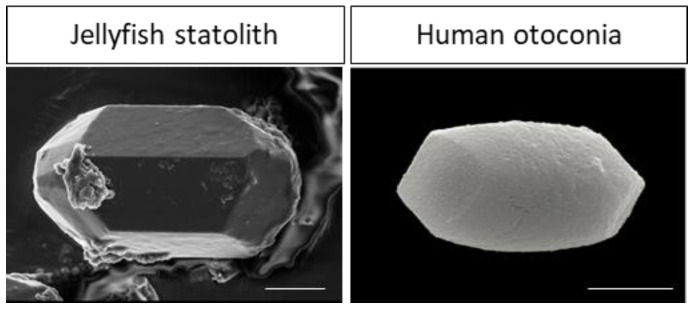
Scanning electron micrographs of statolith from *Sanderia malayensis* (Scyphozoa) and human otoconia. Scale bar = 5 µm. Note that human otoconia and jellyfish statoliths have a similar crystal-shaped appearance. Adapted from [59] for the human otoconia and the statolith micrograph is a generous gift from Dr. Ilka Sötje from University of Hamburg. Scale bar = 5 µm.

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
