# Peer review of "The Vestibular Nuclei: A Cerebral Reservoir of Stem Cells Involved in Balance Function in Normal and Pathological Conditions"

_ijms, 2024, doi:10.3390/ijms25031422_

Round 1
Reviewer 1 Report
Comments and Suggestions for Authors
This is an exceptionally valuable and meaningful review, providing a comprehensive overview of the intriguing realm of neurogenesis in the vestibular nuclei. Moreover, the research team has previously contributed significantly to the field of vestibular studies. I have a few minor questions:
1. Line 283: The author mentions that the UVN model induces neurogenesis in the vestibular nuclei, while the UL model has almost no effect. However, both are classical models for studying vestibular compensation. How do you explain the differences in cell proliferation between these distinct vestibular lesion models?
2. The vestibular nuclei comprise four subnuclei: superior, inferior, medial, and lateral. Which subnucleus is primarily affected by vestibular damage? Is there any related research on this?
3. Line 286: Following acute vestibular injury, a neuroinflammatory response occurs. Is the early neuroinflammatory response beneficial or harmful? Please supplement this study: Previous research has suggested that early anti-inflammatory interventions are disadvantageous for vestibular compensation and mentioned, "These results suggest a beneficial role for acute endogenous neuroinflammation in vestibular compensation." DOI: 10.1186/s12974-021-02222-y
4. Line 341: BDNF is a member of the neurotrophin family and is known to play roles in neuronal survival and differentiation during development. Similar phenomena were observed in the UL model: the study indicates that BDNF and TrkB.FL expression were enhanced, and KCC2 expression was reduced in the ipsi-lesional MVN at 8 h after UL. DOI: 10.1111/febs.13360
Author Response
Reviewer 1
This is an exceptionally valuable and meaningful review, providing a comprehensive overview of the intriguing realm of neurogenesis in the vestibular nuclei. Moreover, the research team has previously contributed significantly to the field of vestibular studies. I have a few minor questions:
- Line 283: The author mentions that the UVN model induces neurogenesis in the vestibular nuclei, while the UL model has almost no effect. However, both are classical models for studying vestibular compensation. How do you explain the differences in cell proliferation between these distinct vestibular lesion models?
This question is pertinent and has been well documented in a work published by our group (Dutheil et al., 2011 doi: 10.1371/journal.pone.0022262). In this study, we investigated the expression of neurogenesis in vestibular nuclei after vestibular lesions of various kinds (nerve VIII section, labyrinthectomy or tetrodotoxin injection). The results show that neurogenesis occurs only after vestibular neurectomy in deafferented vestibular nuclei.
In the UVN model, the severity of the deafferentation that is sudden, total and fast, could induce tissue modifications allowing the cellular microenvironment to switch from a non-neurogenic to a neurogenic area. Indeed, UVN consist of an ablation of both the peripheral vestibular sensors and Scarpa’s ganglion but unilateral labyrinthectomy consist of an ablation of the vestibular sensors alone, with preservation of Scarpa’s ganglion. It appears that one of the key parameters modulating cell proliferation seems to be the integrity of the Scarpa ganglion. Furthermore, it is known that inflammatory responses can participate in cell proliferation, migration, differentiation, survival, and incorporation of newborn cells into neural networks. A strong astro/microglial reaction is observed in the deafferented vestibular nuclei after UVN. This glial reaction is stronger after UVN than after UL.
- The vestibular nuclei comprise four subnuclei: superior, inferior, medial, and lateral. Which subnucleus is primarily affected by vestibular damage? Is there any related research on this?
It's a really interesting question but I wouldn't have a clear answer to it because there isn't much research on the subject. A study in human vestibular nuclei, show that neuronal loss associated with aging occurs in the IVN, the MVN, and the LVN, but not in the superior vestibular nucleus (Alvarez et al., 2000; DOI: https://doi.org/10.1016/S0047-6374(00)00098-1). In the same way, we have also reported that the SVN, which is the structure most involved in oculomotor function, does not present neurogenesis after UVN unlike the lateral, medial or inferior nucleus involved in posturo-locomotor function (Tighilet et al., 2007, doi: 10.1111/j.1460-9568.2006.05267.x.).Based on this study, we can assume that the SVN could be the least impacted nucleus in the event of vestibular damage. To support this hypothesis vestibulo-ocular reflexes are well compensated after unilateral vestibular injuries and, while all vestibular nuclei projects to the oculomotor nuclei, the most important projections come from the SVN (Halassi et al., 2005; https://doi.org/10.1016/j.brainresbull.2005.02.013).
However, concerning the most affected vestibular nucleus, there isn't a clear generalization that one vestibular nucleus is always more affected than others in all cases of vestibular injuries. One a more personal note, the posturo-locomotor symptoms following UVN in rats are so severe and visually impacting that I would tend to say that the vestibulo-spinal bundles originating from the MVN and LVN are strongly impacted and therefore these two nuclei are more severely affected by the lesion.
- Line 286: Following acute vestibular injury, a neuroinflammatory response occurs. Is the early neuroinflammatory response beneficial or harmful? Please supplement this study: Previous research has suggested that early anti-inflammatory interventions are disadvantageous for vestibular compensation and mentioned, "These results suggest a beneficial role for acute endogenous neuroinflammation in vestibular compensation." DOI: 10.1186/s12974-021-02222-y
Recent data support the role of endogenous inflammation following vestibular damage, in regulating neurogenesis and vestibular compensation. Indeed, pharmacological blockade or reinforcement of inflammation during the acute phase of vestibular compensation is accompanied by a worsening of the vestibular syndrome and a delay in vestibular compensation correlated with impaired vestibular neurogenesis (El Mahmoudi et al. 2021; 2022 DOI: 10.3390/cells11172693 and 10.1186/s12974-021-02222-y). However, more studies are needed to confirm these results but also to validate these findings in other models of vestibular damage.
As suggested by the reviewer, we add these two studies and the above paragraph to the corrected manuscript line 328 to 335.
- Line 341: BDNF is a member of the neurotrophin family and is known to play roles in neuronal survival and differentiation during development. Similar phenomena were observed in the UL model: the study indicates that BDNF and TrkB.FL expression were enhanced, and KCC2 expression was reduced in the ipsi-lesional MVN at 8 h after UL. DOI: 10.1111/febs.13360
We would like to thank the reviewer for this pertinent study to include in the manuscript. A few lines have been added to the manuscript on this subject (line 383 to 366).
Reviewer 2 Report
Comments and Suggestions for Authors
This study makes an effort in trying to explore neurogenesis in the vestibular nuclei. However, any revision should be made:
1. Introduction section is very inadequate for a scientific article. Moreover, what their study add to existing literature also unclear.
2. Is there a methodology for selecting the studies included in the review? Please including a Material and methods section in order to clarify the study selection procedure.
Comments on the Quality of English Language
This study makes an effort in trying to explore neurogenesis in the vestibular nuclei. However, any revision should be made:
1. Introduction section is very inadequate for a scientific article. Moreover, what their study add to existing literature also unclear.
2. Is there a methodology for selecting the studies included in the review? Please including a Material and methods section in order to clarify the study selection procedure.
Author Response
This study makes an effort in trying to explore neurogenesis in the vestibular nuclei. However, any revision should be made:
- Introduction section is very inadequate for a scientific article. Moreover, what their study add to existing literature also unclear.
We have modified the introduction to fit the classic format of a scientific article even though this is a review and all the points covered are sufficiently developed in each chapter of the manuscript.
The present review provides a comprehensive overview of the intriguing realm of neurogenesis in the vestibular nuclei. Moreover, our research team has previously contributed significantly to the field of vestibular studies with a particular emphasis on vestibular neurogenesis (Tighilet et al., 2007, 2016; Dutheil et al., 2009, 2011, 2016; Rastoldo et al., 2021, 2022). This study adds to existing literature some valuable questions about vestibular neurogenesis since the discovery of vestibular neurogenesis by our research teams to the demonstration of the presence of neural stem cells in the vestibular nuclei.
- Is there a methodology for selecting the studies included in the review? Please including a Material and methods section in order to clarify the study selection procedure.
As suggested by the reviewer we have included a search strategy and selection criteria in the review (line 51 to 59).